# Entanglement of Arabidopsis Seedlings to a Mesh Substrate under Microgravity Conditions in KIBO on the ISS

**DOI:** 10.3390/plants11070956

**Published:** 2022-03-31

**Authors:** Masataka Nakano, Takuya Furuichi, Masahiro Sokabe, Hidetoshi Iida, Sachiko Yano, Hitoshi Tatsumi

**Affiliations:** 1Department of Biology, Tokyo Gakugei University, 4-1-1 Nukuikita-machi, Koganei-shi 184-8501, Japan; masa4nak@gmail.com (M.N.); iida@u-gakugei.ac.jp (H.I.); 2Research Institute for Science & Technology, Tokyo University of Science, 2641 Yamazaki, Noda-shi 278-8510, Japan; 3Bioscience Core Facility, Research Center for Experimental Modeling of Human Disease, Kanazawa University, 13-1 Takaramachi, Kanazawa-shi 920-8640, Japan; 4Faculty of Human Life Sciences, Hagoromo University of International Studies, 1-89-1 Hamadera-minamimachi, Sakai-shi 592-8344, Japan; tfuruichi@hagoromo.ac.jp; 5Mechanobiology Laboratory, Nagoya University Graduate School of Medicine, 65 Tsurumai, Nagoya-shi 466-8550, Japan; msokabe@med.nagoya-u.ac.jp; 6Human Spaceflight Technology Directorate, Japan Aerospace Exploration Agency, Tsukuba-shi 305-8505, Japan; yano.sachiko@jaxa.jp; 7Department of Applied Bioscience, Kanazawa Institute of Technology (KIT), 3-1 Yatsukaho, Hakusan-shi 924-0838, Japan

**Keywords:** gravity sensing, root entanglement, micro-*G* condition, KIBO on ISS

## Abstract

The International Space Station (ISS) provides a precious opportunity to study plant growth and development under microgravity (micro-*G*) conditions. In this study, four lines of Arabidopsis seeds (wild type, wild-type MCA1-GFP, *mca1*-knockout, and *MCA1*-overexpressed) were cultured on a nylon lace mesh placed on Gelrite-solidified MS-medium in the Japanese experiment module KIBO on the ISS, and the entanglement of roots with the mesh was examined under micro-*G* and 1-*G* conditions. We found that root entanglement with the mesh was enhanced, and root coiling was induced under the micro-*G* condition. This behavior was less pronounced in *mca1*-knockout seedlings, although MCA1-GFP distribution at the root tip of the seedlings was nearly the same in micro-*G*-grown seedlings and the ground control seedlings. Possible involvement of MCA1 in the root entanglement is discussed.

## 1. Introduction

Gravity is one of the environmental cues that direct plant growth and development [1]. Changes in the gravity vector generate a broad range of intracellular signals such as increases in reactive oxygen species [2], cytoplasmic pH in columella cells [3,4], and cytoplasmic calcium ion concentration, [Ca^2+^]_c_ [5,6,7,8]. Among these, the [Ca^2+^]_c_-increase is presumably involved in an early process of the gravitropic response [9,10,11,12,13], and Ca^2+^-permeable mechanosensitive (MS) channels are supposedly involved in the response.

Several MS channel families have been identified in plants, e.g., land plant-specific *mid1*-complementing activity (MCAs) [14,15], the bacterial mechanosensitive channel of small conductance (MscS)-like proteins, MSLs [16,17], two pore potassium channels, TPKs [18], Piezos [19], and reduced hyperosmolality-induced [Ca^2+^]_c_ increase (OSCA) [20]. MCAs, AtTPK4, OSCA, and some MSLs are plant plasma membrane MS cation channels [14,15,20,21,22,23,24,25].

The functional properties of MCA1 have been intensively examined among these MS cation channels; overexpressing MCA1 in the plasma membrane was found to increase Ca^2+^ uptake in Arabidopsis seedlings [16]. Membrane stretching elevated [Ca^2+^]_c_ in Chinese hamster ovary (CHO) cells expressing MCA1, and the expression of Arabidopsis MCA1 was found to lead to enhanced mechanosensitive cation channel activity in the *Xenopus laevis* oocyte plasma membrane [22]. The MCA1 protein synthesized in vitro and reconstituted into liposomal membranes is inherently mechanosensitive and permeable to Ca^2+^ [25]. Under hypergravity conditions, the elongation growth of hypocotyls was suppressed in wild-type Arabidopsis seedlings, whereas the extent of the suppression was reduced in *mca1*-knockout seedlings (*mca1*-KO) but was augmented in *MCA1*-overexpressing seedlings (*MCA1*-OX; driven by 35S promoter of cauliflower mosaic virus) [26,27]. When Arabidopsis seedlings were returned from the upside-down position to the upright position, a very-slow [Ca^2+^]_c_-increase was observed, which was strongly attenuated in *mca1*-KO seedlings, and partially restored in *MCA1*-complemented seedlings, suggesting a possible involvement of MCA1 in gravity sensing [11]. A promoter–GUS assay revealed that MCA1 is expressed in various organs of mature Arabidopsis, including the roots, leaves, stems, flowers, and siliques. Arabidopsis loss-of-function *mca1*-KO mutants have roots that are defective in penetrating a hard agar medium from a soft agar medium, suggesting that they are defective in sensing and/or responding to touch stimuli [16]. These findings suggest that MCA1 is a plasma membrane Ca^2+^-permeable MS cation channel that is potentially involved in gravity sensing and touch sensing in Arabidopsis seedlings.

The International Space Station (ISS) provides a precious opportunity to study root growth and development under the micro-*G* (microgravity) environment [28]. Root coiling under micro-*G* conditions has been reported [29]. In this study, Arabidopsis wild type, *mca1*-KO and *MCA1*-OX seedlings were cultured on a nylon mesh placed on Gelrite-solidified MS-medium in KIBO on the ISS, and root entanglement with the mesh was examined under micro-*G* and 1-*G* conditions. Under the micro-*G* condition, root entanglement with the mesh was enhanced and root coiling was induced. This action was less pronounced in the *mca1*-KO seedlings. This is the first study using WT and *mca1*-KO mutant seedlings on the ISS, and a possible involvement of MCA1 in gravity-dependent root entanglement will be discussed.

## 2. Results

Arabidopsis seedlings were grown on a gel plate containing plant growth medium for 10 days under micro-*G* and centrifugation-generated 1-*G* conditions. A nylon mesh with the seedlings was peeled off from the gel plate and then chemically fixed by an astronaut on the ISS, returned to the earth, then examined for root morphology on the ground four weeks after the collection. Photographic images of the WT seedlings with a nylon mesh are shown in Figure 1. Most of the seedlings grown under micro-*G* gathered on the nylon mesh (Figure 1A), while those grown under artificial 1-*G* were widely dispersed by the post-flight treatment (Figure 1B). The rate of seedlings dispersed in the media was statistically higher in the *mca1*-KO lines than in the WT and *MCA1*-OX lines under the space artificial 1-*G* condition; *MCA1*-overexpressing seedlings were more entangled under micro-*G* than those under the artificial 1-*G* condition, and a similar trend was seen in WT seedlings, but this behavior was hardly observed in any of the lines tested under the micro-*G* condition (Table 1). These results suggest that MCA1 plays an important role in gravity-dependent root entanglement in the mesh.

We noted that part of the roots of the WT seedlings often coiled under the micro-*G* condition (Figure 1D,Ea) but did not coil under artificial 1-*G* (Figure 1Eb). In the WT, *mca1*-KO and *MCA1*-OX lines, straight roots were observed below the mesh with or without coiled roots.

The DIC and confocal fluorescence images of root tips of WT-MCA1-GFP grown under the micro-*G* condition showed no major defect in morphology (Figure 2A–C), and MCA1 was expressed in the plasma membrane at the root tip as reported [16]. On the other hand, cytolysis was often detected in the part closer to the base of the root (e.g., the middle part of the taproots), suggesting the plants were stressed in the micro-*G* environment. The pattern of the Lugol’s iodine staining of the WT seedlings grown under the micro-*G* and 1-*G* environments showed that starch was highly accumulated in the root coils (Figure 2D). The starch content in the root tips (Figure 2D–F), known as one of the important components for the gravity sensing machinery, was not changed.

## 3. Discussion

In this study, we found that the roots of Arabidopsis seedlings got entangled with the mesh more frequently under micro-*G* (10 days), and less frequently entangled under 1-*G* conditions. Under the 1-*G* condition, the percentage of *mca1*-KO seedlings detached from the mesh and dispersed in the solution was significantly higher than that of the WT seedlings, while under the micro-*G* condition, the percentage of *mca1*-KO seedlings detached was not different from those of the other genotypes.

There are many reports on the circumnutation of the aerial parts of flowering plants, but few reports on the circumnutation of roots, e.g., root coils [30]. In these studies, the possible influence of gravity on root coiling has been described, e.g., roots of space-grown seedlings have exhibited a significant difference (compared to the terrestrial controls) in overall growth patterns; for example, they skewed to one direction [31], and root coiling in both WT and the phospholipase A (pPLA-I) mutant increased under micro-*G* conditions [29].

The present study also supports the statement that gravity influences root coiling. Coiled roots were observed only in samples under micro-*G*, and root entanglement with the mesh increased under the micro-*G* compared to the 1-*G* control. These suggest that root coiling can be a possible cause of the entanglement. This could be interpreted as an adaptive morphological change under micro-*G* conditions. We speculated that seedlings are pushed to the gel with considerable force under 1-*G* gravity, but the force must be negligible under micro-*G*. Therefore, seedlings need to strengthen the physical contact to the substrate (gel and the mesh) by increasing the zone of interaction between roots and the gel by root coiling, which results in the root entanglement.

MCA1 is a plasma membrane Ca^2+^-permeable MS channel [25], and is potentially involved in gravity sensing [11,26,27] and touch sensing [15] in Arabidopsis seedlings. The distribution of MCA1-GFP in the root tip of seedlings on the ISS was nearly the same as that of the terrestrial control, supporting that MCA1 is functionally normal. The Lugol’s iodine staining of the WT root also supports that gravity-sensing with the starch statolith system would be established normally under micro-*G*. The touch sensing signal pathway interacts with the columella gravity sensing pathway and may coordinate root growth and coiling [32]. This interaction may explain our results; under 1-*G*, touch sensing and gravity sensing systems work in a coordinated fashion in WTs to entangle the root in the mesh, but in *mca1*-KO seedlings touch sensing does not work in concert with a gravity sensing system to achieve sufficient entanglement, suggesting a possible involvement of MCA1 in the root entanglement under 1-*G*. On the other hand, under micro-*G*, *mca1*-KO and WT seedlings behaved similarly because the gravity sensing system did not work and the touch sensing system (not involving MCA1) solely coordinated root growth and coiling. In soil, downwardly growing roots frequently alter their growth direction to escape obstacles that lie in their paths [33]. This biological response may rely on both touch sensing and gravity sensing. Future studies will elucidate the functional role of MCA1 in this coordinated response.

## 4. Materials and Methods

The materials and methods used in this study are very similar to those used in previous studies [12,26,27], except for the seed sowing process. Briefly, approximately 30 *Arabidopsis thaliana* seeds (Col-0 (WT), WT-MCA1-GFP, *mca1*-KO, *MCA1*-OX) were surface-sterilized, attached to a gamma-ray-sterilized edible paper (Kokko oblate) adhered to the tulle lace (nylon mesh), and dried to prepare the ‘seed paper’ that simplified the sowing process under micro-*G*. The seed papers were stored for two weeks before cultivation on the ISS in the refrigerator until and during the launch to the ISS. The cultivation was initiated by attaching the seed paper to the surface of the gel plate containing plant growth medium (MS medium containing 1× Murashige and Skoog salts, 1% (*w/v*) Sucrose, 0.01% (*w/v*) *myo*-inositol and 0.05% (*w/v*) MES pH 5.8, adjusted with 1 M KOH, solidified with 0.3% Gelrite) in a petri dish (6 cm in diameter). After stratification at 4 °C in the dark for 2 days, seeds were incubated at 22 °C in a Plant Experiment Unit (PEU) growth chamber, under continuous light (the ratio of red and blue LED light was 4:1, 80 μM/m^2^/s) for 10 days under micro-*G* and centrifugation that generated artificial 1-*G* conditions in the Cell Biology Experiment Facility (CBEF) unit [34] in KIBO on the ISS; 1-*G* was applied from direction of the stem to the root, as shown in Figure 1A.

On day 10, seedlings were collected by peeling off the nylon mesh from the growth medium. Samples were divided into two groups; one group (WT-MCA1-GFP) was fixed with a fixing solution (4% paraformaldehyde, 0.1% glutaraldehyde, pH7.2) by using Chemical Fixation Apparatus (CFA) at room temperature for 1 day, and stored at 4 °C until observation; the other (WT, WT-MCA1-GFP, *mca1*-KO, *MCA1*-OX) was submerged in RNAlater™ Stabilization Solution (AM7020, Thermo Fisher Scientific) using a CFB unit [35] as shown in Figure 1A and stored at −80 °C until observation. The set of experimental materials was carried to the ISS via SpaceX CRS-8 (8–10 April 2016). These samples were observed 28 days after fixation on the ground. One plate was analyzed for each genotype and condition.

Seedlings on the mesh were inserted into the solution-containing apparatus for RNA-stabilization on the ISS and were returned to the earth within the apparatus. Seedling were gently taken out from the apparatus and placed in a 10 cm culture dish. The experimental protocol was repeated for WT, *mca1*-KO, *MCA1*-OX, and WT-MCA1-GFP seedlings. During this process, some seedlings remained on the mesh and others were dispersed. The number of seedlings on the mesh (i.e., entangled seedlings) and suspended in the medium (i.e., dispersed seedlings) were counted and/or photographed. Confocal imaging of MCA1-GFP expressing roots was made with a Zeiss LSM-410 or Leica DMI-6000 microscope. The roots of the WT seedlings were stained with Lugol’s iodine solution and observed with a binocular stereomicroscope. Significant differences in the proportions of samples were evaluated using the software Origin (proportion test, Origin Lab). An RNA assay was not included in this study.

## Figures and Tables

**Figure 1 plants-11-00956-f001:**
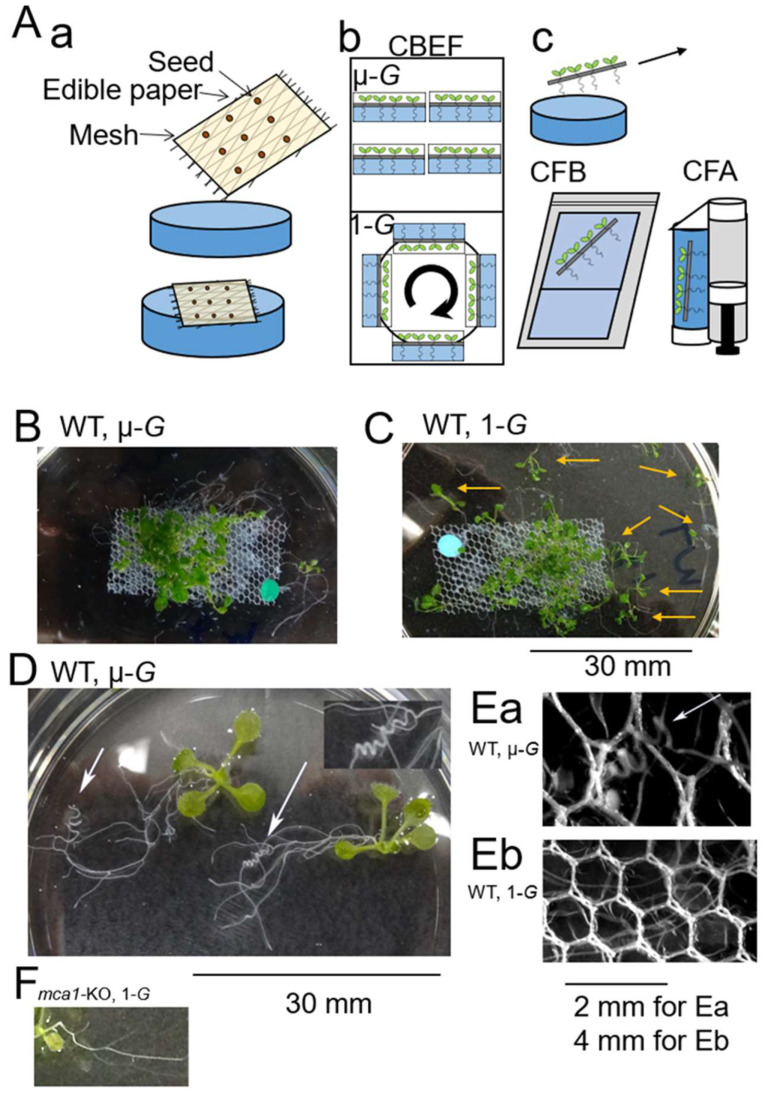
Seedlings entangled with a mesh under different conditions. (**A**) Schematic illustration of experiments. (**a**) Seeds were attached to an edible paper (oblate) adhered to a nylon mesh on the earth, and the ‘seed paper’ was attached to the surface of the gel plate in a petri dish on the ISS; an oblate polymer made with starch was dissolved on the gel plate, the cultivation was initiated, and seedlings were grown on the gel plate. (**b**) Arabidopsis seedlings were grown on the gel plate under micro-*G* (upper chambers of CBEF) and centrifugation-generated 1-*G* conditions from the stem to the root direction (lower chamber of CBEF; cell biology experiment facility) as in the case of the gravitational force on the earth’s surface, equipped with PEU (plant experiment unit, a small growth chamber for plants). (**c**) Seedlings were collected by peeling off the nylon mesh from the gel plate and inserted to an apparatus for RNA-stabilization (CFB unit) and for paraformaldehyde (CFA unit), and seedlings were returned to the earth and examined. Plants were submerged in the fixative (blue) inside the plastic bag (CFB; chemical fixation bag) or inside the cylinder (CFA; chemical fixation apparatus). The fixative was introduced by sliding a piston in CFA. (**B**) Seedlings grown under micro-*G* conditions on the ISS. (**C**) Seedlings under 1-*G* conditions generated by centrifugation on the ISS. Arrows indicate the dispersed seedlings. (**D**) Two typical seedlings under micro-*G* conditions without mesh. Arrows indicate the root coils. The region shown by the right arrow is magnified (2.5 times) (upper inset). Two seedlings dispersed from the mesh are imaged to show the coiled roots. The insert in panel (**D**) shows the magnified root coil. (**E**) Roots along with the mesh. (**Ea**) A root coil (shown by the arrow) found in a space micro-*G* plant; (**Eb**) straight and curved roots were found in space 1-*G* seedlings. (**F**) An image of *mca1*-KO seedlings with roots that do not coil under 1-*G* on ISS. Images of wild-type seedlings are shown in panels (**B**–**E**) and *mca1*-KO seedlings in (**F**). Images in (**D**,**F**) share the same scale bar.

**Figure 2 plants-11-00956-f002:**
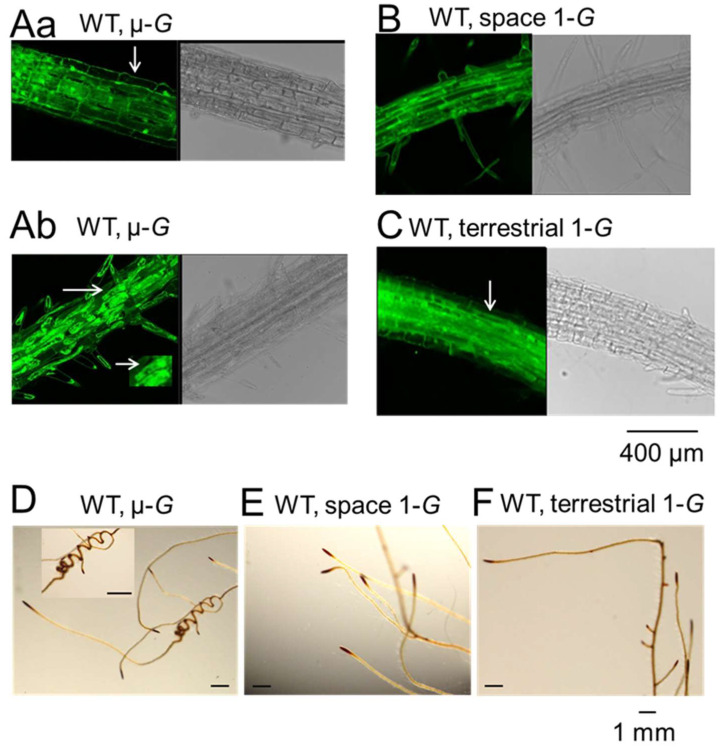
Confocal images and Lugol’s iodine staining of WT-MCA1-GFP in roots. (**Aa**) A sample under micro-*G* conditions; near the root tip (upper panel **Aa**) and a part closer to the base of the root (lower panel **Ab**); (**B**) under artificial 1-*G* conditions generated by centrifugation in ISS; C, under 1-*G* conditions on the terrestrial conditions. The arrow in (**Aa**) (upper panel) shows the plasma membrane, and that in the lower panel (**Ab**) shows cytolysis (majority of cells show cytolysis), and that in (**C**) shows the plasma membrane. The root in panel (**B**) was a few centimeters from the root tip. Dissociation of the cell membrane from the cell wall was used as a sign of cytolysis. Typical cell cytolysis is shown in the inset of (**Ab**) (see the arrow). The pattern of the Lugol’s iodine staining of the WT root under micro-*G* (**D**) and 1-*G* (**E**) terrestrial 1-*G* (**F**). The root coil magnified is shown in the inset (**D**).

**Table 1 plants-11-00956-t001:** Plants entangled with a mesh and those dispersed in the medium under different conditions. Letters in the table column denote significant differences between the groups; statistical significance was seen between “a” and “b”, “b” and “d” (*p* < 0.01), and “c” and “b” (*p* < 0.05) (the two-sample proportion test). Data of the *MCA1*-OX and WT-MCA1-GFP lines were gathered and analyzed as *MCA1*-overexpressing seedlings because both the *MCA1* and *MCA1-GFP* genes were expressed from the same promoter, 35 S CaMV; the proportions of seedlings dispersed under micro-*G* (3/39, 7.7%) and 1-*G* (27/53, 50.9%) are statistically different (*p* < 0.01). Under the micro-*G* condition, the percentage of *mca1*-KO seedlings detached (11.7%) was nearly the same as that of the WT (13%) and *MCA1*-OX seedlings (4%). There is no statistical difference. The proportion of the number of WT-MCA1-GFP seedlings dispersed under micro-*G* was (2/14, 14%) and 1-*G* was (21/28, 75%). These data show the same tendency as for the other seedlings. WT-MCA1-GFP seedlings are not included in the Table 1, since a certain fraction of WT-MCA1-GFP seedlings were fixed with paraformaldehyde (PFA). On the other hand, WT, *mca1*-KO, and *MCA1*-OX seedlings were treated with RNA-stabilizer.

	Space Micro-*G*		Space Artificial 1-*G*	
	Seedlings on Mesh	Seedlings Dispersed	Seedlings on Mesh	Seedlings Dispersed
WT	20	3 (13%)	20	11 (35%) “c”
*mca1*-KO	15	2 (11.7%) “a”	3	17 (85%) “b”
*MCA1*-OX	24	1 (4%)	19	6 (24%) “d”

## Data Availability

Not applicable.

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
