# Peer review of "Entanglement of Arabidopsis Seedlings to a Mesh Substrate under Microgravity Conditions in KIBO on the ISS"

_plants, 2022, doi:10.3390/plants11070956_

Round 1
Reviewer 1 Report
The study by Nakano et al. presents some novel insights on plant root gravitropism using a micro-gravity experimental condition achieved by performing experiments on ISS. It shows the morphology of Arabidopsis thaliana roots in micro-gravity and suggests the role of mechanosensitive Ca2+-ion channel MCA1 in acquiring this morphology by examining mca1- knockout, and MCA1-overexpressing plants.
The main methods of comparison of seedlings are 1) evaluation of seedling dispersal out of the nylon mesh placed on Gelrite substrate and 2) visual detection of coils on roots. Although the study seems to be rigorous and methods correct at a first glance, there are several concerns that can be either addressed textually, or by adding data to which authors refer, but don’t present in the current manuscript.
Major comments:
1) The main results claimed by authors in the abstract are not substantiated in the text by data and even formulated oppositely in the discussion:
The main result is formulated in the abstract as follows: “We found that root entanglement with the mesh was enhanced, and root coiling was induced under the micro-G condition. This behavior was less pronounced in mca1-knockout seedlings” (line 25)
Unfortunately, there is now image of mca1-knockout seedlings in Figures to see if this is true.
Besides that, in Discussion we can read a somewhat opposite statement: “while under the micro- G condition, the percentage of mca1-KO seedlings detached was not different from those of the other genotypes” (line 128). The data to substantiate this conclusion exists in Table 1, however there is no statistical comparison done to make this conclusion.
2) In Table 1 there is no data presented on MCA1-OX and WT MCA1-GFP, although the caption contains information on it.
3) There is no information about the number of plates (biological replicates) in the study. From the numbers presented in Table 1 it looks like only 1 plate was analyzed for each genotype & condition. Please add information about number of replicates and number of seedlings analyzed in each replicate.
4) There is no information of how 1G is experimentally achieved and in which direction. Also there is not enough details to understand at which timepoint the solidification of agar substrate occurs. Unfortunately, without all this information the reader can suspect, that 1G was applied when seeds didn’t have roots yet and substrate was not solid enough, so that seeds can float out of the mesh and disperse. Seeds didn’t disperse in micro-gravity condition just because there is no force or substrate movement that makes them float. Please describe the procedure more rigorously to eliminate this suspicion. It is nevertheless hard to imagine how the seedlings can move out of the mesh if agar was solid from the beginning of the experiment. Which forces make them move? Provide the proof that such forces are the same for all specimens, in micro-G and in 1G conditions.
I suggest that authors revise the manuscript according to these suggestions and add data to support each of their conclusion.
Minor comments:
Introduction
Add more recent review about gravitropism if possible (in introduction). Add references to the studies conducted on ISS. If it is the very first study on ISS, please mention it.
Explain abbreviation “CHO” cells. (line 45)
Line 64: “development in this unique environment.” – write directly how unique, is it only micro-gravity? (what is the value of G there?), no other differences to your knowledge/ experimental settings?
Line 66: “root entanglement with the mesh was examined under micro-G and 1-G conditions”. Add literature ref why this aspect is examined, is it standard? Is it possible to compare this parameter “entanglement” to other studies?
Results
Line 79: “MCA1-OX lines under the space artificial 1-G condition; MCA1-overexpressing”
Add info about promoter and where it is expressed.
Table 1 - please add MCA1-OX data.
Better to present this data as a graph and add to Fig.1. Table is also useful, but can be moved to Supplementary Material.
Line 103 “(Fig. 1C)” - should be (Fig. 1C and Fig.1Da)
Line 103 “In all types, straight..”. Should be genotypes if I understood the message correctly. Otherwise rephrase.
Line 111 “highly accumulated in the root coils. (Fig. 2D and E)”. There are no coils in E and it was said before that under 1G roots don’t coil. Change Figure citing. Does this result fit to published works on coils? Add refs on that.
Figures
In general: indicate conditions (microG / 1G) and genotypes directly on Fig panels for easier reading.
Fig.1A,B: add arrows which show exactly where to look to see the difference between two conditions.
Fig.1B: - a light blick in the middle – please replace by better image if the roots on the mesh where used for evaluation of entanglement
Fig.1C: doesn’t contain the mesh. Please explain the reason and purpose of it in the text.
Fig.1C: image for 1G should be added for comparison, to see that at 1G roots never coil.
Fig.1Da Db - images should be in the same scale to be comparable. If desired different zoom (2x zoom for example) can be added in addition:
Fig.2B Add information on which part of the root it is. Is it comparable to Aa or Ab?
Fig2 A,B,C should have same orientation, it is difficult to compare like this
Fig2 (Ab) shows cytolysis (majority of cells show cytolysis) – arrow hardly visible, change its position/orientation to not obscure the cells. What are exactly signs of cytolysis?
Discussion
Line 125: Add something like “In this study we have shown that…” to the first paragraph to clearly state what is shown in this study.
Line 137: use the word “statement” instead of “idea”
Line 139: “root coiling is another aspect of the entanglement” – not clear sentence. What is the first aspect of entanglement? Please rewrite to make it clear.
Materials and Methods
Describe the procedure of how 1G is achieved, in which direction?
Add method of evaluation/quantification of entanglement and dispersal of seedlings.
Author Response
Reviewer Comments:
Reviewer 1
The study by Nakano et al. presents some novel insights on plant root gravitropism using a micro-gravity experimental condition achieved by performing experiments on ISS. It shows the morphology of Arabidopsis thaliana roots in micro-gravity and suggests the role of mechanosensitive Ca2+-ion channel MCA1 in acquiring this morphology by examining mca1- knockout, and MCA1-overexpressing plants.
The main methods of comparison of seedlings are 1) evaluation of seedling dispersal out of the nylon mesh placed on Gelrite substrate and 2) visual detection of coils on roots. Although the study seems to be rigorous and methods correct at a first glance, there are several concerns that can be either addressed textually, or by adding data to which authors refer, but don’t present in the current manuscript.
Major comments:
Comment (1) The main results claimed by authors in the abstract are not substantiated in the text by data and even formulated oppositely in the discussion:
The main result is formulated in the abstract as follows: “We found that root entanglement with the mesh was enhanced, and root coiling was induced under the micro-G condition. This behavior was less pronounced in mca1-knockout seedlings” (line 25)
Comment (1−1) Unfortunately, there is now image of mca1-knockout seedlings in Figures to see if this is true.
Comment (1−2) Besides that, in Discussion we can read a somewhat opposite statement: “while under the micro- G condition, the percentage of mca1-KO seedlings detached was not different from those of the other genotypes” (line 128). The data to substantiate this conclusion exists in Table 1, however there is no statistical comparison done to make this conclusion.
Reply (1−1) An image of mca1-knockout seedlings under the 1- G condition is shown in figure 1F. We added text in the legend of figure 1 as follows; lines 110-111, “F, an image of mca1-KO seedlings with roots that do not coil under 1-G on ISS.”
Reply (1−2) Under the micro-G condition, the percentage of mca1-KO seedlings detached (11.7%) was nearly the same as that of the WT (13%) and MCA1-overexpressing seedlings (4%). There is no statistical difference. We described the statistical comparison in the legend of Table 1 as follows; lines 118-120, “Under the micro-G condition, the percentage of mca1-KO seedlings detached (11.7%) was nearly the same as that of the WT (13%) and MCA1-OX seedlings (4%). There is no statistical difference.”
Comment (2) In Table 1 there is no data presented on MCA1-OX and WT MCA1-GFP, although the caption contains information on it.
Reply (2) We added information of the number of MCA1-OX in the new Table 1, because WT, mca1-KO, and MCA1-OX seedlings were treated in the same way in the space experiment. These set of data were analyzed statistically. We added data of WT-MCA1-GFP seedlings to that of MCA1-OX because both the MCA1 and MCA1-GFP genes were expressed from the same promoter, 35 S CaMV and MCA1-OX and WT-MCA1-GFP seedlings behaved in a similar way in space micro-G and space artificial 1-G. However, a certain fraction of WT-MCA1-GFP seedlings were treated slightly differently (fixed with paraformaldehyde instead of RNA-stabilizer). This assembly of data is necessary to show that the percentage of MCA1-OX seedlings on the mesh (i.e., entangled seedling) is significantly larger under space micro-G than the percentage under space artificial 1-G. This assembly is needed due to the limitation in the number of seeds allowed to be sent to the ISS. Due to these we did not add information of WT-MCA1-GFP seedlings to the Table 1. The legend of new Table 1 is changed as follows; lines 120-125, “The proportion of the number of WT-MCA1-GFP seedlings dispersed under micro-G is (2/14, 14%) and 1-G is (21/28, 75%). These data show the same tendency as for the other seedlings. WT-MCA1-GFP seedlings was not included in the Table 1, since a certain fraction of WT-MCA1-GFP seedlings were fixed with paraformaldehyde with PFA. On the other hand, WT, mca1-KO, and MCA1-OX seedlings were treated with RNA-stabilizer with PFB mechanical, with different mechanical fluid mechanics.”
Comment (3) There is no information about the number of plates (biological replicates) in the study. From the numbers presented in Table 1 it looks like only 1 plate was analyzed for each genotype & condition. Please add information about number of replicates and number of seedlings analyzed in each replicate.
Reply (3) One plate was analyzed for each genotype & condition, and one plate contained 20 – 30 seedlings. We mentioned these in the text as follows; line 218-219 “One plate was analyzed for each genotype and condition.”
Comment (4) There is no information of how 1G is experimentally achieved and in which direction. Also, there is not enough details to understand at which timepoint the solidification of agar substrate occurs. Unfortunately, without all this information the reader can suspect, that 1G was applied when seeds didn’t have roots yet and substrate was not solid enough, so that seeds can float out of the mesh and disperse. Seeds didn’t disperse in micro-gravity condition just because there is no force or substrate movement that makes them float. Please describe the procedure more rigorously to eliminate this suspicion. It is nevertheless hard to imagine how the seedlings can move out of the mesh if agar was solid from the beginning of the experiment. Which forces make them move? Provide the proof that such forces are the same for all specimens, in micro-G and in 1G conditions.
Reply (4) Arabidopsis seedlings were grown on the gel plate containing plant growth medium for 10 days under micro-G and centrifugation-generated 1-G conditions from the stem to the root, which was perpendicular to the surface of the gel plate. The timepoint the solidification of gel substrate occurs several weeks before the SpaceX launch. We added illustrations to describe the procedure in Fig. 1, and schematically illustrates the protocol of experiments as follows; (1) on the earth Arabidopsis thaliana seeds were attached to an edible paper (oblate) adhered to a nylon mesh, (2) on the ISS this ‘seed paper’ is attached to the surface of the gel plate in a petri dish (6 cm in diameter), (3) an oblate polymer, made with starch was dissolved on the gel plate, the cultivation was initiated, and Arabidopsis seedlings were grown on the gel plate, (4) seedlings were collected by peeling off the nylon mesh from the gel plate and inserted to an apparatus for RNA-stabilization (or fixation), (5) seedlings were returned to the earth and examined. We mentioned these in the legend of figure 1 as follows; lines 91-103 “A, Schematic illustration of experiments. (a) Seeds were attached to an edible paper (oblate) adhered to a nylon mesh on the earth, and the ‘seed paper’ was attached to the surface of the gel plate in a petri dish on the ISS; an oblate polymer, made with starch was dissolved on the gel plate; the cultivation was initiated; and seedlings were grown on the gel plate. (b) Arabidopsis seedlings were grown on the gel plate under micro-G (upper chambers of CBEF; cell biology experiment facility) and centrifugation-generated 1-G conditions from the stem to the root direction (lower chamber of CBEF) as in the case of gravitational force on the earth's surface, equipped with PEU (plant experiment unit, an small growth chamber for plants). (c) Seedlings were collected by peeling off the nylon mesh from the gel plate and inserted to an apparatus for RNA-stabilization (CFB unit) and for paraformaldehyde (CFA unit), and seedlings were returned to the earth and examined. Plants were submerged in the fixative (blue) inside the plastic bag (CFB; chemical fixation bag) or inside the cylinder (CFA; chemical fixation apparatus). The fixative was introduced by sliding a piston in CFA.”
Seedling moved out of the mesh during the process of fixation on the ISS and transporting from the ISS to the earth. We described this in the materials and methods as follows; lines 220-226 “Seedlings on the mesh were inserted into the solution-containing apparatus for RNA-stabilization on the ISS, and were returned to the earth within the apparatus. Seedling were gently taken out from the apparatus, and place it in a 10 cm-culture dish. The experimental protocol was repeated for WT, mca1-KO, MCA1-OX, and WT-MCA1-GFP seedlings. During this process, some seedlings remained on the mesh and others were dispersed. The number of seedlings on the mesh (i.e., entangled seedlings) and suspended in the medium (i.e., dispersed seedlings) were counted and/or photographed.”
Comment (5) I suggest that authors revise the manuscript according to these suggestions and add data to support each of their conclusion.
Reply (5) We revised the manuscript according to the reviewer`s suggestions.
Minor comments:
Introduction
Comment (1) Add more recent review about gravitropism if possible (in introduction). Add references to the studies conducted on ISS. If it is the very first study on ISS, please mention it.
Reply (1) We did not find a recent review about gravitropism of plants conduced on the ISS between 2015 to 2022. This study is the very first study using WT and mca1-KO seedlings on the ISS. We mentioned this in the text as follows; lines 72-74 “This is the first study using WT and mca1-KO mutant seedlings on the ISS, and a possible involvement of MCA1 in gravity-dependent root entanglement will be discussed.”
Comment (2) Explain abbreviation “CHO” cells. (line 45)
Reply (2) Chinese hamster ovary (CHO) cells (line 45-46)
Comment (3) Line 64: “development in this unique environment.” – write directly how unique, is it only micro-gravity? (what is the value of G there?), no other differences to your knowledge/ experimental settings?
Reply (3) We inserted a word “micro-G” as follows; Line 66, “development under the micro-G (microgravity) environment.”
Comment (4) Line 66: “root entanglement with the mesh was examined under micro-G and 1-G conditions”. Add literature ref why this aspect is examined, is it standard? Is it possible to compare this parameter “entanglement” to other studies?
Reply (4) We added a literature that root coiling increased under micro-G conditions in the introduction as follows; lines 66-67 “Root coiling under micro-G conditions is reported [29].”
Results
Comment (5) Line 79: “MCA1-OX lines under the space artificial 1-G condition; MCA1-overexpressing”
Add info about promoter and where it is expressed.
Reply (5) We added info about promoter and location of expression as follows; line 53 “(MCA1-OX; driven by 35S promoter of cauliflower mosaic virus)” line 58- 59 “MCA1 is expressed in various organs of mature Arabidopsis, including the roots, leaves, stems, flowers, and siliques.”
Comment (6) Table 1 - please add MCA1-OX data. Better to present this data as a graph and add to Fig.1. Table is also useful, but can be moved to Supplementary Material.
Reply (6) We added MCA1-OX data. The numbers of seedlings on the mesh in different experimental conditions is one of the main results of this study. Due to the limitation of the space experiment, the number of samples is relatively small, it may not be appropriate to display these data graphically. Therefore, we would like to keep the " Table " in the main text.
Comment (7) Line 103 “(Fig. 1C)” - should be (Fig. 1C and Fig.1Da)
Reply (7) Thank you for this. We replaced “(Fig. 1D and Fig.1Ea)” in the “new” Fig. 1 as suggested; see line 128.
Comment (8) Line 103 “In all types, straight.”. Should be genotypes if I understood the message correctly. Otherwise rephrase.
Reply (8) We use genotypes as suggested as follows; lines 128-129 “In the WT, mca1-KO and MCA1-OX lines”
Comment (9) Line 111 “highly accumulated in the root coils. (Fig. 2D and E)”. There are no coils in E and it was said before that under 1G roots don’t coil. Change Figure citing. Does this result fit to published works on coils? Add refs on that.
Reply (9) Thank you. We changed figure citing as follows; line 138 “The pattern of the Lugol's iodine staining of the WT seedlings grown under the micro-G showed that starch was highly accumulated in the root coils (Fig. 2D).” This result does not fit to published works, and we did not add references on that.
Figures
Comment (10) In general: indicate conditions (microG / 1G) and genotypes directly on Fig panels for easier reading.
Reply (10) We indicated conditions, (micro-G / 1-G) and genotypes, directly in the panels in Fig. 1 and Fig. 2.
Comment (11) Fig.1A,B: add arrows which show exactly where to look to see the difference between two conditions.
Reply (11) We add arrows to point the scattered seedlings in panel C in the “new Fig. 1”, and explained in the legend as follows; lines 105 “Arrows indicate the dispersed seedlings.”
Comment (12) Fig.1B: - a light blick in the middle – please replace by better image if the roots on the mesh where used for evaluation of entanglement
Reply (12) We placed a new image of the same sample in panel C of the new Fig. 1.
Comment (13) Fig.1C: doesn’t contain the mesh. Please explain the reason and purpose of it in the text.
Reply (13) The seedlings grown under artificial 1-G were dispersed during the insertion to the apparatus for RNA-stabilization (and fixation) and the travel from the ISS to earth. We chose two seedlings which were dispersed and roots were coiled as shown in Fig. 1C in order to show more clearly the morphology of the coiled root without mesh. We explained this in the figure legend as follows; line 105-106 “Two seedlings dispersed from the mesh were imaged to show the coiled roots.”
Comment (14) Fig.1C: image for 1G should be added for comparison, to see that at 1G roots never coil.
Reply (14) We added image of roots never coil at 1-G (lower inset); figure legend line 108-111 “The insert in panel D shows the magnified root coil. E, Roots along with the mesh; (a) A root coil (shown by the arrow) found in a space micro-G plant, (b) Straight and curved roots were found in space 1-G seedlings. F, an image of mca1-KO seedlings with roots that do not coil under 1-G on ISS. Images of wild-type seedlings are shown in panels B-E and mca1-KO seedlings in F.”
Comment (15) Fig.1Da Db - images should be in the same scale to be comparable. If desired different zoom (2x zoom for example) can be added in addition:
Reply (15) The image in panel Ea shows a coiled root. Higher magnification is needed to show the coils. On the other hand, panel Eb needs to show long straight roots, so it requires low magnification. Therefore, we want to keep these panels as they are.
Comment (16) Fig.2B Add information on which part of the root it is. Is it comparable to Aa or Ab?
Reply (16) The root in panel B was a few centimeters from the tip. It may comparable to Aa or C. We mentioned this in the figure legend as follows; lines147-148 “The root in panel B was a few centimeters from the root tip and its position was comparable to those shown in panel Aa and C.”
Comment (17) Fig2 A, B, C should have same orientation, it is difficult to compare like this
Reply (17) We have arranged them as suggested; see new Fig. 2.
Comment (18) Fig2 (Ab) shows cytolysis (majority of cells show cytolysis) – arrow hardly visible, change its position/orientation to not obscure the cells. What are exactly signs of cytolysis?
Reply (18) We have arranged the arrow. Dissociation of the cell membrane from the cell wall was used as a sign of cell lysis. We magnified the cell cytolysis and is shown in the inset of Ab. We mentioned this in the figure legend as follows; lines148-150 “Dissociation of the cell membrane from the cell wall was used as a sign of cytolysis. Typical cell cytolysis is shown in the inset of Ab (see the arrow).”
Discussion
Comment (19) Line 125: Add something like “In this study we have shown that…” to the first paragraph to clearly state what is shown in this study.
Reply (19) Thank you for this. We added words as suggested; line 154 “In this study, we have shown that “
Comment (20) Line 137: use the word “statement” instead of “idea”
Reply (20) We replaced idea to statement as follows; line 166 “The present study also supports the statement that ”
Comment (21) Line 139: “root coiling is another aspect of the entanglement” – not clear sentence. What is the first aspect of entanglement? Please rewrite to make it clear.
Reply (21) We agree that it is not clear, and we rewrite it as follows; lines 167-170 “The present study also supports the statement that gravity influences root coiling. Coiled roots were observed only in samples under micro-G, and root entanglement with the mesh increased under the micro-G compared to the 1-G control. These suggest that root coiling can be a possible cause of the entanglement. This could be interpreted as an adaptive morphological change under micro-G conditions.”
Materials and Methods
Comment (22) Describe the procedure of how 1G is achieved, in which direction?
Reply (22) We mentioned the direction as follows; lines 208-209“;1-G was applied from the stem to the root direction as shown in Fig. 1A.” We describe the procedure of how 1-G is achieved in Fig. 1A and the legend.
Comment (23) Add method of evaluation/quantification of entanglement and dispersal of seedlings.
Reply (23) We mentioned the method in the text as follows; lines 220-226, “Seedlings on the mesh were inserted into the solution-containing apparatus for RNA-stabilization on the ISS, and were returned to earth within the apparatus. Seedling were gently taken out from the apparatus, and place it in a 10 cm-culture dish. The experimental protocol was repeated for WT, mca1-KO, MCA1-OX, and WT-MCA1-GFP seedlings. During this process, some seedlings remained on the mesh and others were dispersed. The number of seedlings on the mesh (i.e., entangled seedlings) and suspended in the medium (i.e., dispersed seedlings) were counted and/or photographed.”

Reviewer 2 Report
This manuscript by Masataka Nakano et al. (PLANTS #.......: "Entanglement of Arabidopsis Seedlings to a Mesh Substrate under Microgravity Conditions on the ISS") reports on an analysis of root entanglement on a mesh substrate under microgravity conditions on ISS and 1 g control conditions obtained by centrifugation on ISS, for wild type, MCA1-knockout, MCA1- and MCA1-GFP overexpressing Arabidopsis thaliana seedlings. The results show an increased ability for wild type and over-expressing roots to coil and get entangled within the mesh substrate under microgravity relative to 1-g control conditions. Interestingly, these behaviors are attenuated in mca1-knockout and MCA1-over-expressing seedlings even though MCA1-GFP expression in the root tip is similar between microgravity and 1-g control conditions. The authors also show that starch accumulates in root coils whereas starch content within the root tip is not altered between conditions. Finally, analysis of MCA1-GFP expression indicates increased cytolysis in the roots exposed to microgravity relative to 1g controls, suggesting stress responses. The authors conclude from these data that MCA1 may contribute to the modulation of root coiling and entanglement through its effect on gravity sensing (which is minimal under microgravity) and touch sensing. Overall, the data are novel and interesting. However, the scarcity of experimental method description largely prevents me from truly evaluating the quality of these data and their interpretation. Important points to address in this manuscript include:
- What is the genotype of the seedlings shown in Figure 1?
- Figure 2 shows an image of a root tip subjected to a 1-G control under terrestrial conditions. Yet, none of the data reported in this manuscript (other than this one) include this important control.
- In the 1-G centrifugation control on ISS, how were the plates positioned relative to the axis of centrifugation? Did this affect the ability of the roots to grow straight through the mesh into the gelled medium without entanglement?
- If seeds were positioned on the mesh at the top of the gelled medium at the time of experiment activation on ISS, how did they become loose on the plate? They should remain in place while germinating (they do not move by themselves). Is the dispersion occurring as a consequence of fixative application at the end of the experiment?
- Directly relevant to point 3, how was fixation carried out? Was the fixative directly applied to the plate? This information is not clearly presented in Materials and Methods.
- Also, when were the pictures shown in Figure 1 taken: Right after growth before transfer to fixative, or after transfer to fixative?
- The evidence for cytolysis under microgravity illustrated in figure 2 is not obvious to me. Could higher-resolution images (close-ups) be provided to illustrate this point?
- Starch staining of the root tip under microgravity and 1-G control conditions should be shown at similar resolutions (and higher magnification) to truly compare the levels of staining. The current images suggest less staining under microgravity than in the 1-G control, but this could be a consequence of the different levels of magnification used in these two images.
- Table 2 should include the data from both MCA1- and MCA1-GFP-overexpressing lines separately, as part of the table, and the statistical analysis should compare each one of those lines (as well as the total of both, as currently succinctly shown in the legend) to the WT and mca1-KO under microgravity and 1G conditions.
Author Response
Reviewer Comments:
Reviewer 2
This manuscript by Masataka Nakano et al. (PLANTS #.......: "Entanglement of Arabidopsis Seedlings to a Mesh Substrate under Microgravity Conditions on the ISS") reports on an analysis of root entanglement on a mesh substrate under microgravity conditions on ISS and 1 g control conditions obtained by centrifugation on ISS, for wild type, MCA1-knockout, MCA1- and MCA1-GFP overexpressing Arabidopsis thaliana seedlings. The results show an increased ability for wild type and over-expressing roots to coil and get entangled within the mesh substrate under microgravity relative to 1-g control conditions. Interestingly, these behaviors are attenuated in mca1-knockout and MCA1-over-expressing seedlings even though MCA1-GFP expression in the root tip is similar between microgravity and 1-g control conditions. The authors also show that starch accumulates in root coils whereas starch content within the root tip is not altered between conditions. Finally, analysis of MCA1-GFP expression indicates increased cytolysis in the roots exposed to microgravity relative to 1g controls, suggesting stress responses. The authors conclude from these data that MCA1 may contribute to the modulation of root coiling and entanglement through its effect on gravity sensing (which is minimal under microgravity) and touch sensing. Overall, the data are novel and interesting. However, the scarcity of experimental method description largely prevents me from truly evaluating the quality of these data and their interpretation. Important points to address in this manuscript include:
(Comment 1) What is the genotype of the seedlings shown in Figure 1?
Reply (1) Wild type and mca1-KO seedlings are shown in figure 1. We mentioned it in the legend of figure 1 as follows; lines 103-111 “B, Seedlings grown under micro-G conditions on the ISS. C, Seedlings under 1-G conditions generated by centrifugation on the ISS. Arrows indicate the dispersed seedlings. D, Two typical seedlings under micro-G conditions without mesh. Arrows indicate the root coils. The region shown by the right arrow is magnified (2.5 times) (upper inset). Two seedlings dispersed from the mesh are imaged to show the coiled roots. The insert in panel D shows the magnified root coil. E, Roots along with the mesh; (a) A root coil (shown by the arrow) found in a space micro-G plant, (b) Straight and curved roots were found in space 1-G seedlings. F, an image of mca1-KO seedlings with roots that do not coil under 1-G on ISS. Images of wild-type seedlings are shown in panels B-E and mca1-KO seedlings in F.”
(Comment 2) Figure 2 shows an image of a root tip subjected to a 1-G control under terrestrial conditions. Yet, none of the data reported in this manuscript (other than this one) include this important control.
Reply (2) Thank you for pointing this. We added the control results under terrestrial conditions in figure 2F; legend of figure 2, line 151 “The pattern of the Lugol's iodine staining of the WT root under micro-G (D), 1-G (E) and terrestrial 1-G (F).”
(Comment 3) In the 1-G centrifugation control on ISS, how were the plates positioned relative to the axis of centrifugation? Did this affect the ability of the roots to grow straight through the mesh into the gelled medium without entanglement?
Reply (3) On ISS Arabidopsis seedlings were grown on the gel plate containing plant growth medium. The surface of the plate was positioned perpendicular to the direction of centrifugal force as illustrated in figure 1A. The direction of the centrifugation-generated 1-G was perpendicular to the surface of the gel plate. In other words, the 1-G centrifugal force was applied in the direction from the stem to the root as in the gravitational force on earth. Arabidopsis thaliana seeds were attached to an edible paper (oblate) adhered to a nylon mesh, this ‘seed paper’ was attached to the surface of the wettable gel plate, the oblate polymer was dissolved in the gel plate and seeds were attached to the gel, the cultivation was initiated, and Arabidopsis seedlings were grown on the gel plate as illustrated in figure 1A. Thus, most of the seeds attached directly to the gel, and the cultivation was initiated. These suggest that the 1-G centrifugal force does not affect the ability of the roots to grow straight through the mesh into the gelled medium, since the roots probably grow under the 1-G centrifugal force as in the case where the 1-G gravitational force on the earth`s surface. Roots of Arabidopsis presumably grow without entanglement as observed in many labs. We mentioned these in the legend of figure 1 as follows; lines 91-99 “A, Schematic illustration of experiments. (a) Seeds were attached to an edible paper (oblate) adhered to a nylon mesh on the earth, and the ‘seed paper’ was attached to the surface of the gel plate in a petri dish on the ISS; an oblate polymer, made with starch was dissolved on the gel plate; the cultivation was initiated; and seedlings were grown on the gel plate. (b) Arabidopsis seedlings were grown on the gel plate under micro-G (upper chambers of CBEF; cell biology experiment facility) and centrifugation-generated 1-G conditions from the stem to the root direction (lower chamber of CBEF) as in the case of gravitational force on the earth's surface, equipped with PEU (plant experiment unit, an small growth chamber for plants).”
(Comment 4) If seeds were positioned on the mesh at the top of the gelled medium at the time of experiment activation on ISS, how did they become loose on the plate? They should remain in place while germinating (they do not move by themselves). Is the dispersion occurring as a consequence of fixative application at the end of the experiment?
Reply (4) Seeds were positioned on the edible paper (oblate) not on the mesh. The seed and the paper were attached to the surface of the wettable gel plate. The oblate polymer was dissolved in the gel plate, and cultivation of seeds was initiated. See the protocol in figure 1 and figure legend. The dispersion of seedlings from the mesh presumably occurred as a consequence of fixative application at the end of the experiment due to the insertion of seedlings to the apparatus for fixation (CFA and CFB), and presumably during the transportation from the ISS to the earth. We mentioned these in the materials and methods as follows; lines 220-226 “Seedlings on the mesh were inserted into the solution-containing apparatus for RNA-stabilization on the ISS, and were returned to the earth within the apparatus. Seedling were gently taken out from the apparatus, and place it in a 10 cm-culture dish. The experimental protocol was repeated for WT, mca1-KO, MCA1-OX, and WT-MCA1-GFP seedlings. During this process, some seedlings remained on the mesh and others were dispersed. The number of seedlings on the mesh (i.e., entangled seedlings) and suspended in the medium (i.e., dispersed seedlings) were counted and/or photographed.”, and figure 1 legend; lines 91-95 “A, Schematic illustration of experiments. (a) Seeds were attached to an edible paper (oblate) adhered to a nylon mesh on the earth, and the ‘seed paper’ was attached to the surface of the gel plate in a petri dish on the ISS; an oblate polymer, made with starch was dissolved on the gel plate; the cultivation was initiated; and seedlings were grown on the gel plate.”
(Comment 5) Directly relevant to point 3, how was fixation carried out? Was the fixative directly applied to the plate? This information is not clearly presented in Materials and Methods.
Reply (5) Fixative or RNA-stabilization solution was applied to the seedlings at the end of the experiment by inserting the seedlings to the apparatus for fixation with forceps as illustrated in figure 1, and explained in the Figure 1 legend about the apparatus for fixation (CFA and CFB). We cited reference 34 and 35 in the revised text, which gives more information.
(Comment 6) Also, when were the pictures shown in Figure 1 taken: Right after growth before transfer to fixative, or after transfer to fixative?
Reply (6) The pictures shown in Figure 1 was taken after transfer to fixative (or RNA-stabilization solution) and seedlings returned to the earth. We mentioned this in the figure legend of Fig. 1 as follows; lines 99-101 “(c) Seedlings were collected by peeling off the nylon mesh from the gel plate and inserted to an apparatus for RNA-stabilization (CFB unit) and for paraformaldehyde (CFA unit), and seedlings were returned to the earth and examined.”
(Comment 7) The evidence for cytolysis under microgravity illustrated in figure 2 is not obvious to me. Could higher-resolution images (close-ups) be provided to illustrate this point?
Reply (7) Dissociation of the cell membrane from the cell wall was used as a sign of cytolysis. We magnified the typical cytolysis and put it in the inset in the figure 2Ab and explained in the figure legend as follows; lines 148-150 “Dissociation of the cell membrane from the cell wall was used as a sign of cytolysis. Typical cell cytolysis is shown in the inset of Ab (see the arrow).”
(Comment 8) Starch staining of the root tip under microgravity and 1-G control conditions should be shown at similar resolutions (and higher magnification) to truly compare the levels of staining. The current images suggest less staining under microgravity than in the 1-G control, but this could be a consequence of the different levels of magnification used in these two images.
Reply (8) We checked the magnification and contrast of the images and fixed them. Unfortunately, we do not have images with higher magnification. When we took photo of these samples, we examined the sample with an optical microscope but did not realize the subtle difference in the starch staining as reviewer suggested. We would like to leave the text as it is.
(Comment 9) Table 2 should include the data from both MCA1- and MCA1-GFP-overexpressing lines separately, as part of the table, and the statistical analysis should compare each one of those lines (as well as the total of both, as currently succinctly shown in the legend) to the WT and mca1-KO under microgravity and 1G conditions.
Reply (9) We added information of the number of MCA1-OX in the new Table 1, because WT, mca1-KO, and MCA1-OX seedlings were treated in the same way in the space experiment. These set of data were statistical analyzed. We added data of WT-MCA1-GFP seedlings to that of MCA1-OX because both the MCA1 and MCA1-GFP genes were expressed from the same promoter, 35 S CaMV and MCA1-OX and WT-MCA1-GFP seedlings behaved in a similar way in space micro-G and space artificial 1-G. However, a fraction of WT-MCA1-GFP seedlings were treated slightly differently (fixed with paraformaldehyde); on the other hand, WT, mca1-KO, and MCA1-OX seedlings were treated with RNA-stabilizer. This assembly of data is required to show that the percentage of MCA1-OX seedlings on the mesh (i.e., entangled seedlings) is significantly larger under space micro-G than that under space artificial 1-G. This assembly is needed due to the limitation in the number of seeds allowed to be sent to the ISS. For these reasons data from WT-MCA1-GFP seedling are not included in the Table, but we mentioned them in the legend. The legend of new Table 1 is changed as follows; lines 120-125 “The proportion of the number of WT-MCA1-GFP seedlings dispersed under micro-G is (2/14, 14%) and 1-G is (21/28, 75%). These data show the same tendency as for the other seedlings. WT-MCA1-GFP seedlings was not included in the Table 1, since a certain fraction of WT-MCA1-GFP seedlings were fixed with paraformaldehyde with PFA. On the other hand, WT, mca1-KO, and MCA1-OX seedlings were treated with RNA-stabilizer with PFB mechanical, with different mechanical fluid mechanics.”

Round 2
Reviewer 2 Report
This manuscript describes the effect of microgravity on ISS on Arabidopsis seedling root entanglement in a nylon mesh relative to a 1-G centrifugation control. This process is accompanied by strong root coiling under microgravity for wild type and MCA1 over-expressing seedlings, and it is affected by a knockout mutation in the MCA1 gene (which encodes a mechanosensitive ion channel previously implicated in gravity and touch responses. The authors conclude that MCA1 may contribute to the modulation of root coiling and entanglement through its effect on gravity sensing (which is minimal under microgravity) and touch sensing. As mentioned in my review of the original draft of this paper, the data are novel and interesting, providing importanmt new information with potential application in the development of plants for use in bioregenerative life-support systems usable in space exploration missions.
The authors have nicely addressed all the comments and suggestions I made on the previous draft of the manuscript, better detailing the experimental approach used in this project. In particular, the changes made to figure 1 are very helpful, as are the additional experimental details added to the text. I only have very minor suggestions for improvement of this interesting document:
- In line 82, I would add "by the post-flight treatment" at the end of the sentence to emphasize that seedling dispersal is a consequence of the post-growth treatment. This sentence would now read as follows: " (...) while those grown under artificial 1-G were widely dispersed (Fig. 1B)".
- In the legend to Table 1, line 125, "with PFA" should probably be replaced by PFA within parentheses. Also, in lines 126-127, the final section of the sentence (", with different mechanical fluid mechanics") is hard to understand (duplication of mechanics). Would something like " with different mechanical fluid dynamics" be more appropriate?
- Lines 149-150 (legend to Figure 2). It seems that the root in panel B is actually not really at a position comparable to those roots shown in panels Aa and C. Indeed, the root in panel B appears to have fully developed root hairs whereas those in panels Aa and C seem to be where root hairs are initiated (panel Aa) or are undergoing growth (panel C). This sentence should be corrected.
Author Response
Comment 1. In line 82, I would add "by the post-flight treatment" at the end of the sentence to emphasize that seedling dispersal is a consequence of the post-growth treatment. This sentence would now read as follows: " (...) while those grown under artificial 1-G were widely dispersed (Fig. 1B)".
Reply 1. We added "by the post-flight treatment" at the end of the sentence. Line 83. Thank you for this.
Comment 2. In the legend to Table 1, line 125, "with PFA" should probably be replaced by PFA within parentheses. Also, in lines 126-127, the final section of the sentence (", with different mechanical fluid mechanics") is hard to understand (duplication of mechanics). Would something like " with different mechanical fluid dynamics" be more appropriate?
Reply 2. We fixed the sentence; we put PFA with parentheses, and removed “with PFB mechanical, with different mechanical fluid mechanics.” Lines 125-126.
Comment 3. Lines 149-150 (legend to Figure 2). It seems that the root in panel B is actually not really at a position comparable to those roots shown in panels Aa and C. Indeed, the root in panel B appears to have fully developed root hairs whereas those in panels Aa and C seem to be where root hairs are initiated (panel Aa) or are undergoing growth (panel C). This sentence should be corrected.
Reply 3. Thank you for pointing out this. We removed “and its position was comparable to those shown in panel Aa and C.” Line 148.
